# Solution-Processed Silicon Doped Tin Oxide Thin Films and Thin-Film Transistors Based on Tetraethyl Orthosilicate

**DOI:** 10.3390/membranes12060590

**Published:** 2022-06-01

**Authors:** Ziyan He, Xu Zhang, Xiaoqin Wei, Dongxiang Luo, Honglong Ning, Qiannan Ye, Renxu Wu, Yao Guo, Rihui Yao, Junbiao Peng

**Affiliations:** 1State Key Laboratory of Luminescent Materials and Devices, Institute of Polymer Optoelectronic Materials and Devices, South China University of Technology, Guangzhou 510640, China; 201930171079@mail.scut.edu.cn (Z.H.); 201921020680@mail.scut.edu.cn (X.Z.); msyeqiannan@mail.scut.edu.cn (Q.Y.); 201930175046@mail.scut.edu.cn (R.W.); 201930170485@mail.scut.edu.cn (Y.G.); psjbpeng@scut.edu.cn (J.P.); 2Southwest Institute of Technology and Engineering, Chongqing 400039, China; weixiaoqin810913@163.com; 3Guangzhou Key Laboratory for Clean Energy and Materials, Huangpu Hydrogen Innovation Center, School of Chemistry and Chemical Engineering, Institute of Clean Energy and Materials, Guangzhou University, Guangzhou 510006, China; luodx@gdut.edu.cn

**Keywords:** tetraethyl orthosilicate, tin oxide, thin-film transistors, oxygen vacancy

## Abstract

Recently, tin oxide (SnO_2_) has been the preferred thin film material for semiconductor devices such as thin-film transistors (TFTs) due to its low cost, non-toxicity, and superior electrical performance. However, the high oxygen vacancy (V_O_) concentration leads to poor performance of SnO_2_ thin films and devices. In this paper, with tetraethyl orthosilicate (TEOS) as the Si source, which can decompose to release heat and supply energy when annealing, Si doped SnO_2_ (STO) films and inverted staggered STO TFTs were successfully fabricated by a solution method. An XPS analysis showed that Si doping can effectively inhibit the formation of V_O_, thus reducing the carrier concentration and improving the quality of SnO_2_ films. In addition, the heat released from TEOS can modestly lower the preparation temperature of STO films. By optimizing the annealing temperature and Si doping content, 350 °C annealed STO TFTs with 5 at.% Si exhibited the best device performance: I_off_ was as low as 10^−10^ A, I_on_/I_off_ reached a magnitude of 10^4^, and V_on_ was 1.51 V. Utilizing TEOS as an Si source has a certain reference significance for solution-processed metal oxide thin films in the future.

## 1. Introduction

In recent years, due to their high mobility, low temperature preparation, and compatibility with flexible processes, metal oxide semiconductor (MOS) materials represented by indium gallium zinc oxide (IGZO) have been extensively applied in flat panel displays such as AMLCD and AMOLED, driven by TFTs [1,2,3,4,5]. However, the scarce reserve of indium in the earth’s crust (0.25 ppm) leads to its high market price (~$750/kg) [6]. Furthermore, it is toxic, which makes it incompatible with the trend of the consumer electronics market toward low cost environmental benignity. The development of an alternative indium-free oxide semiconductor material system is imperative. Notably, the electronic structure of Sn^4+^ (4d^10^5s^2^) is similar to that of In^3+^ (4d^10^5s^0^) with the spherical symmetry s orbit, leading to the high mobility of SnO_2_ and In_2_O_3_ even in an amorphous state [7,8]. In addition, Sn is abundant (2.2 ppm) and relatively inexpensive (~$15/kg) [6]. SnO_2_ is also non-toxic, environmentally friendly, and chemically stable, making it the most promising candidate to replace In-based MOS materials in semiconductor devices such as TFTs.

SnO_2_-based TFTs have generally been fabricated by magnetron sputtering and other vacuum technologies [9,10,11,12], but those involve an expensive, complex process dependent on a vacuum environment. In contrast, the solution method has broad development prospects in modern electronic device processing [13,14,15], with the advantages of low cost and a simple process ofmanipulation by doping. As a result, solution-processed SnO_2_-TFTs have increasingly become a preferred method.

Oxygen vacancy (V_O_) plays a significant role in carrier concentration, and then affects the properties of the material [16,17]. In 2010, Tsay et al. [18] prepared crystalline SnO_2_ thin films at 500 °C by spin coating, with the O/Sn ratio of only 1.69 and a carrier concentration of 7.5 × 10^18^ cm^−3^ due to the existence of V_O_. An excess of carriers caused by a high concentration of V_O_ in SnO_2_ leads to TFT performance deterioration, including a large off current (I_off_) and difficulty in turning off [19,20]. Many studies have been conducted to suppress the V_O_ concentration by doping. In 2020, Zhang et al. [21] prepared Ga doped SnO_2_–TFT (GTO–TFT) at 450 °C by spin coating, and found that with the Ga content rising from 20% to 60%, the V_O_ decreased from 30.24% to 17.18%, while the I_off_ of TFT correspondingly decreased from 10^−3^ A to 10^−11^ A. In addition, other commonly used dopants such as Sb, Cr, Zr, Y [22,23,24,25,26] can also reduce the V_O_ concentration, but low reserves and a certain toxicity limit their practical application.

However, Si is environmentally friendly, non-toxic, and resource-rich. Si^4+^ has the same valence state as Sn^4+^, and will not introduce new charges into SnO_2_. In addition, the binding energy of Si–O (799.6 kJ/mol) is higher than that of Sn–O (531.8 kJ/mol), and the Lewis acid strength of Si (8.096) is also significantly higher than that of Sn (1.617), which makes Si a superior oxygen binder to suppress the formation of V_O_ [27,28,29]. Liu et al. [30] fabricated silicon doped SnO_2_–TFTs (STO–TFTs) by sputtering, controlling the V_O_ concentration with Si, and the best device performance was obtained with 1 wt.% Si: the saturation mobility (μ_sat)_ was 6.38 cm^2^/(V·s), the on/off current ratio (I_on_/I_off_) was 1.44 × 10^7^, and the subthreshold swing (SS) was 0.77 V/Dec. Therefore, incorporating Si has the potential to lower the carrier concentration of SnO_2_ films and improve the device’s performance. However, there are few studies of Si doping into SnO_2_ by the solution method, and most of them require a high processing temperature (>450 °C) [26,31].

Considering the above problems, this paper utilized tetraethyl orthosilicate (TEOS) and tin chloride dihydrate (SnCl_2_·2H_2_O) to prepare STO thin films, and the effects of Si doping content on the chemical composition, microstructure, and electrical properties of SnO_2_ were investigated. It was found that TEOS can not only act as an Si dopant to diminish the V_O_ and carrier concentrations, but can also modestly reduce the preparation temperature of SnO_2_ thin films due to its decomposition and heat release when annealing. In a previous study, it was demonstrated that the AlO_x_: Nd film is a suitable dielectric in oxide TFTs due to its high dielectric constant and low leakage current density [32]. Based on this, bottom-gate and top-contact STO TFTs were successfully fabricated.

## 2. Materials and Methods

0.1 mol/L SnO_2_ precursor solutions were synthesized by dissolving SnCl_2_·2H_2_O in 2-methoxyethanol (2-ME), followed by stirring for 0.5 h to mix well. TEOS was added at an atomic ratio (Si/Sn at.%) of 2.5, 5, 10, and 15, respectively. Before spin coating, the precursor solutions were stirred for 12 h in the air. Figure 1 shows the preparation process of the STO films. The alkali free glass substrate was treated with oxygen plasma with a power of 60 W for 10 min. 40 μL solutions filtered through a 0.22 μm syringe filter were added dropwise to glass substrate, and then spun by a spin coater at 5000 rpm for 30 s to prepare SnO_2_ and STO wet films. The resulting films were transferred to a hot plate heated at 100 °C for 10 min to evaporate the organic solvent, followed by annealing at 300 °C for 1 h to obtain dense films.

The TFTs were fabricated with a bottom-gate and top-contact configuration, as shown in Figure 2. The preparation process for the active layer was essentially the same as that shown in Figure 1, except that the substrates were composed of Al: Nd/Al_2_O_3_: Nd (the thickness of Al: Nd gate electrode was 200 nm and Al_2_O_3_: Nd insulator was 300 nm with a capacitance per unit area of 38 nF/cm^2^), the Si doping concentrations were 0, 2.5, and 5 at.%, the spin speed was 8000 rpm, and the annealing temperatures were 300 °C and 350 °C. The S/D electrodes were deposited on the surface of the STO films by direct current (DC) sputtering of an Al target with a purity of 99.99%. The sputtering power was 100 W with a deposition pressure of 1 mTorr and a time of 1200 s. The patterning of electrodes was realized by masking the non-S/D electrode area, with a channel width of 800 μm and length of 200 μm (W:L=800:200).

The thermal characteristics of the precursors were measured with a thermogravimetric analyzer (TG) (DZ-TGA101, Nanjing Shelley biology, Nanjing, China) and a differential scanning calorimeter (DSC) (DZ-DSC300C, Nanjing Shelley biology, Nanjing, China) at a heating rate of 10°C/min from room temperature to 500 °C under ambient conditions. The contact angles of the solutions were tested by an optical contact angle meter (Biolin, Theta Lite 200, Gothenburg, Sweden). The surface morphology of STO films were observed with laser scanning confocal microscopy (LSCM) (OLS50-CB, Tokyo, Japan) and an atomic force microscope (AFM) (BY 3000, Being Nano-Instruments, Guangzhou, China). The microstructure of the STO thin films was characterized by an X-ray diffractometer (XRD) (PANalytical Empyrean, Almelo, The Netherlands). Microwave photoconductivity decay (μ-PCD) (KOBELCO, LTA-1620SP, Kobe, Japan) was performed to clarify the distribution of internal defects in the films. The electrical parameters of the STO films were obtained by Hall (ECOPIA, HMS 5300, Seoul, Korea) measurement. The chemical compositions were analyzed by X-ray photoelectron spectroscopy (XPS) (Thermo Fisher Scientific, Nexsa, MA, USA). A semiconductor parameter analyzer (Primarius FS-Pro, Shanghai, China) was employed to measure the electrical characteristics of the TFTs.

## 3. Results

Figure 3 shows the STO precursors with varying Si doping content after stirring for 12 h. The pure SnO_2_ precursor is colorless and transparent without precipitation, indicating that SnCl_2_·2H_2_O had been fully dissolved in 2-ME, which is conducive to improving the uniformity of the film. After adding TEOS, the precursor displays no obvious change, implying that TEOS has better solubility in the solvent, and Si is evenly dispersed in the precursor.

Figure 4a shows the DSC-TG curves of SnO_2_ precursors with 0, 2.5, and 5 at.% Si. For 0 at.% Si, the mass ratio declines rapidly from 99% to 14% during 20~147 °C, with a significant endothermic peak at 133.5 °C. The main process in this stage is the large evaporation of 2-ME (boiling point: 124.5 °C) and sol–gel reaction of Sn^2+^ [33]. The temperature continues to rise, but the mass decreases slowly, corresponding to the gradual removal of impurities and the conversion of SnO_2_. After 341.7 °C, no obvious weight loss was observed, suggesting that SnO_2_ has been completely transformed. Equations (1)–(3) show the reaction process [34,35]. The thermal behavior of an STO precursor with 5 at.% Si is similar to that of 0 at.% Si, but its endothermic peak of solvent evaporating shifts to 114.4 °C.
(1)$${\mathrm{SnCl}}_{2}\cdot 2H_{2}O+2{\mathrm{CH}}_{3}{\mathrm{OCH}}_{2}{\mathrm{CH}}_{2}\mathrm{OH}\to \mathrm{Sn}{\left({\mathrm{OCH}}_{2}{\mathrm{CH}}_{2}{\mathrm{OCH}}_{3}\right)}_{2}+2H_{2}O+2\mathrm{HCl}↑$$
(2)$$2\mathrm{Sn}{\left({\mathrm{OCH}}_{2}{\mathrm{CH}}_{2}{\mathrm{OCH}}_{3}\right)}_{2}+6H_{2}O+O_{2}\to 2\mathrm{Sn}{\left(\mathrm{OH}\right)}_{4}+4{\mathrm{CH}}_{3}{\mathrm{OCH}}_{2}{\mathrm{CH}}_{2}\mathrm{OH}↑$$
(3)$$\mathrm{Sn}{\left(\mathrm{OH}\right)}_{4}\to {\mathrm{SnO}}_{2}+2H_{2}O↑$$

Figure 4b displays the local enlarged view of the TG curves for further comparison. It was found that TEOS can markedly reduce the conversion temperature of SnO_2_, which shifts toward a lower temperature with the increase of Si doping content. The complete conversion temperatures of SnO_2_ in different STO precursors are 341.7 °C (0 at.% Si), 227.7 °C (2.5 at.% Si), and 130.2 °C (5 at.% Si). After the complete transformation of SnO_2_, the continued rising temperature can promote the diffusion of O in air into the STO films, which partially compensates for the V_O_ and reduces the carrier concentration, as shown in Figure 4c.

The above phenomena are ascribed to the decomposition and heat release of TEOS during high-temperature annealing, which provides more energy for film formation [36,37]. Absorbing the extra energy from TEOS prompts the endothermic peak of the evaporating solvent to shift toward a lower temperature, and promotes the formation of O-Sn-O, as shown in Figure 5, which can modestly reduce the preparation temperature of SnO_2_ films. 

In order to study the wettability of STO precursors on the substrate surface, the contact angle of precursors on the alkali free glass was tested, with the results shown in Figure 6. It was found that the contact angle of a pure SnO_2_ precursor on glass substrate is relatively low (16.15°), indicating decent contact on the substrates. After adding Si, the contact angle of an STO solution on the substrate decreases, as low as 9.82° when doping 10 at.% Si. This demonstrates that Si doping can improve the wettability of SnO_2_ precursor solution on the substrate surface, which is conducive to improving the quality of films. Good wettability can reduce the interface defects between the film and the substrate surface, and ensure the successful progress of spin coating preparation and device manufacturing.

LSCM was employed to obtain the surface morphology of 300 °C annealed STO films, and the captured microphotographs are displayed in Figure 7a. It can be observed that all STO films are flat and uniform in large scale without physical defects such as holes and cracks, while white particles appear on the surface of pure SnO_2_ film, indicating that adding Si is beneficial for improving the film quality.

The surface roughness of thin films affects the interface contact and the device performance. Figure 7b shows the AFM 3D images of STO films with a scanning area of 10 × 10 μm^2^. The root mean square (Sq) of STO films is generally lower than that of pure SnO_2_ film, indicating that Si can reduce the surface roughness, which is consistent with the LSCM. The Sq of the STO film with 2.5 at.% Si is as low as 0.23 nm, and, with the rising Si content, the Sq slightly increases to 0.34 nm. Its smooth surface is conducive to decreasing the density of interface defects and subsequently improving the device performance.

Figure 8 shows the XRD patterns of STO films with different Si concentrations. It was found that the STO films with 0 at.% and 2.5 at.% Si are amorphous. When the Si concentration increases to 5 at.%, crystallization peaks occur at 26.63°, 33.83°, and 52.13°, respectively corresponding to the diffraction peaks of SnO_2_ on the (110), (101), and (211) crystal planes [22]. Furthermore, XRD patterns reveal no Si element-related diffraction peaks even with 10 at.% Si, implying that there is no obvious second phase in the films and SnO_2_ remains the main component. In addition, as Si increases from 5 at.% to 10 at.%, the diffraction peaks of SnO_2_ become sharper, representing enhanced crystallinity. This can be attributed to the increased exothermic heat and energy supply with the rising of TEOS content. However, for 15 at.% Si, the diffraction peaks disappear completely, which may be explained by a large amount of Si entering into the SnO_2_ crystal, destroying its normal lattice structure, and, thus, suppressing the crystallization of SnO_2_.

The internal defects of the film significantly affect the carrier concentration of the film and the performance of devices. Figure 9 shows the results of a μ-PCD test. The τ_2_ is correlated to the recombination rate of photogenerated carriers in the film. Shallow level defects can trap photogenerated carriers, thus reducing the recombination rate. The larger the mean peak and τ_2_, the higher the shallow level defect density rises [38,39,40]. Figure 9 shows that, compared with 0 at.% Si, the mean peak value of the STO film with 2.5 at.% Si declines markedly from 26.10 mV to 6.70 mV, and τ_2_ value decreases from 2.04 μs to 0.42 μs. This suggests that 2.5 at.% Si doping can effectively diminish the density of shallow level defects in SnO_2_ films, which is conducive to lowering the carrier concentration of the films. However, as Si content increases from 2.5 at.% to 15 at.%, the peak value and τ_2_ increase gradually, revealing that a high content of Si can increase the density of shallow level defects in SnO_2_. Singhal et al. [41] reported the same trend that doping Co increases defect content in TiO_2_. The variation of defects in the semiconductor material is ascribed to the shift of Fermi level when doping, which can result in spontaneous formation of the compensating charged defects [42].

Figure 10 displays the O 1s XPS spectra of STO thin films as a function of Si concentration. The O elements in SnO_2_ films mainly exist in the form of lattice oxygen (L_O_), adsorbed oxygen (A_O_), and V_O_. The high density of V_O_ is the dominant factor for high carrier concentration of SnO_2_ films. The characteristic peaks of O can be deconvoluted into three peaks by their different binding energy (L_O_: ~530 eV, A_O_: ~532 eV, V_O_: ~531 eV [43]). The variation of L_O_/(L_O_ + V_O_ + A_O_) and V_O_/(L_O_ + V_O_ + A_O_) with the content of Si can be calculated by Gaussian fitting. In particular, the area under the V_O_ peak is proportional to the concentration of oxygen vacancy, which acts as defects as well as electron donors [16,17,44]. Compared with 0 at.% Si, the V_O_ ratio of STO film with 2.5 at.% Si decreases remarkably from 29.78% to 16.69%, as seen in Figure 10, indicating that Si can effectively suppress V_O_ and reduce the carrier concentration. Meanwhile, the L_O_ ratio increases substantially from 59.38% to 83.31%, implying that the addition of Si can induce the formation of O–Sn–O and preserve its structure [45]. However, as the Si concentration rises from 2.5 at.% to 15 at.%, the V_O_ ratio in STO films slightly increases, but is still lower than 0 at.% Si. This may be due to a disordered structure whereby a large amount of Si is intercalated in the lattice [40], as indicated by the L_O_ ratio (Figure 10f). Consequently, the density of V_O_ can be regulated by varying the Si doping content, and the control of carrier concentration in the SnO_2_ film can be realized.

The electrical properties of the active layer are critical factors for TFT performance. Figure 11a shows the Hall test results of STO films with different Si concentrations. With the increase in Si content, sheet carrier concentration first decreases and then increases, which is in line with the variation trend of the peak value, τ_2_, and V_O_ ratio with Si concentration. This indicates that the addition of Si affects the electrical properties of STO films by regulating internal defect density such as V_O_. Compared with 0 at.% Si, the sheet carrier concentration of the STO film with 5 at.% Si declines from 2.19 × 10^14^ cm^−2^ to 5.84 × 10^13^ cm^−2^, implying that Si doping can effectively diminish the carrier concentration of SnO_2_. In addition, it was observed that with the increased content of Si, although the sheet carrier concentration of STO films is lower than that of pure SnO_2_ film, the hall mobility of STO films gradually decreases, which can most likely be attributed to the scattering caused by the enhanced crystallization, as concluded in the XRD analysis.

In order to devise a suitable Si concentration range for the preparation of TFTs, I–V curves of STO films with 0, 2.5, and 5 at.% Si were investigated under the condition of a 5 V working voltage, as shown in Figure 11b. The response currents of STO films with 0, 2.5, and 5 at.% Si were 3.49 × 10^−9^ A, 3.76 × 10^−10^ A, and 2.34 × 10^−9^ A, respectively. This phenomenon shows that STO films with 2.5 at.% Si have the potential to reduce the I_off_ of TFTs.

Based on previous analyses, it was found that STO films with 2.5 at.% Si showed better properties, such as the lowest V_O_ ratio of 16.69% and a response current of 3.76 × 10^−10^ A at 5 V. Therefore, STO TFTs with 2.5 at.% Si were further fabricated with an annealing temperature of 300 °C and 350 °C. Their transfer characteristics were measured under the conditions of V_GS_ = ±30 V and V_DS_ = 20.1 V, as shown in Figure 12a. The following performance parameters of corresponding STO TFTs were extracted: on/off current ratio (I_off_/I_off_), off current (I_off_), the subthreshold swing (SS) of 300 °C annealed TFT of 3.46 × 10^3^, 7.74 × 10^−9^A, and 5.50 V/Dec, respectively; and that of 350 °C annealed TFT of 7.43 × 10^3^, 1.19 × 10^−9^ A, and 4.24 V/Dec, respectively. Compared with 300 °C annealing, the STO TFT fabricated at 350 °C has a higher I_off_/I_off_, a lower I_off_, and a smaller SS. The decrease of I_off_ is probably a result of the increasing temperature that promotes the compensation of V_O_ in the films, and then reduces carrier concentration, as analyzed in Figure 4c. Simultaneously, the rising temperature allows SnO_2_ to obtain enough energy for the internal structure to reorganize and diminish the defect density at the interface between the STO film and the Al_2_O_3_: Nd dielectric layer, leading to the reduction of the SS. However, the mobility (μ_sat_) of 350 °C annealed STO TFT (0.32 cm^2^/(V·s)) is lower than that at 300 °C (0.81 cm^2^/(V·s)), which may be attributed to the enhanced crystallinity of STO films, and, thus, the μ_sat_ degrades with the increased scattering caused by the grain boundary [46].

Since the device prepared at 350 °C shows better performance, 350 °C annealed STO TFTs with 0, 2.5, and 5 at.% Si were further fabricated. The transfer characteristics obtained are shown in Figure 12b, and all devices exhibit good switching characteristics. Table 1 shows the extracted performance parameters of corresponding TFTs. As the Si content rises from 0 at.% to 5 at.%, it was found that (1) I_off_ gradually declines while I_on_/I_off_ gradually increases, indicating that Si doping can effectively suppress the formation of V_O_, thus reducing the carrier concentration of the active layers of the STO TFT; (2) Voltage corresponding to the TFT switching from an off state to an on state (V_on_) gradually decreases, which is conducive to lowering power consumption in practical applications; and (3) the SS gradually reduces, probably due to the increased heat release caused by the rising concentration of TEOS, which is conducive to the reorganization of SnO_2_ and subsequent reduction in internal defect states. After optimization, the 350 °C annealed STO TFT with 5 at.% Si exhibits the best performance, with a μ_sat_ of 0.13 cm^2^/(V·s), I_off_ of 2.01 × 10^−10^ A, I_on_/I_off_ of 1.04 × 10^4^, V_on_ of 1.51 V, and SS of 3.48 V/Dec.

## 4. Conclusions

In this paper, STO TFTs were fabricated by spin coating with TEOS as an Si dopant, and the effects of Si doping concentrations on the properties of SnO_2_ were explored. During annealing, TEOS can decompose to release heat and supply energy for film formation, which is helpful to appropriately reduce the preparation temperature of the film and improve its quality. With the rising of Si content, the increased exothermic heat of TEOS led to the enhanced crystallization of the STO films, while excessive Si can destroy the lattice and degrade the crystallinity. In addition, Si doping can effectively suppress the V_O_ concentration, and the V_O_ ratio of 2.5 at.% Si doped STO film was as low as 16.69%. The variation trends of a shallow level defect density, V_O_ ratio, and carrier concentration were concurrent with the change in Si concentration, which first decreased and then increased, indicating that Si doping regulates the electrical properties of the film by controlling defect states such as V_O_. Following optimization, it was confirmed that 350 °C annealed and 5 at.% Si doped STO TFT showed the best performance, as I_off_, I_on_/I_off_, and V_on_ were 2.01 × 10^−10^ A, 1.04 × 10^4^, and 1.51 V, respectively. These developments offer a foundation for further study of MOS-based films and devices prepared by the solution method.

## Figures and Tables

**Figure 1 membranes-12-00590-f001:**
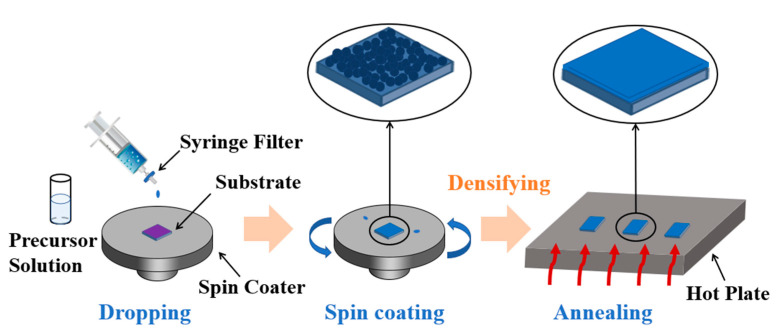
Sketch map of STO thin film process.

**Figure 2 membranes-12-00590-f002:**
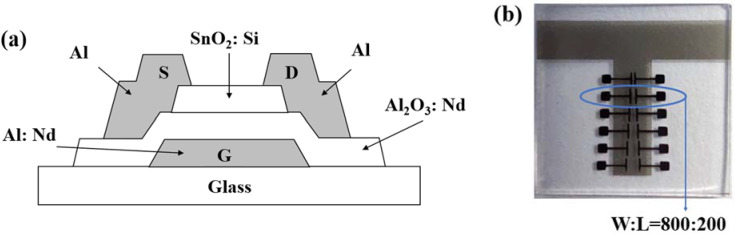
(**a**) Sketch map of STO TFT and (**b**) photo of STO TFT.

**Figure 3 membranes-12-00590-f003:**
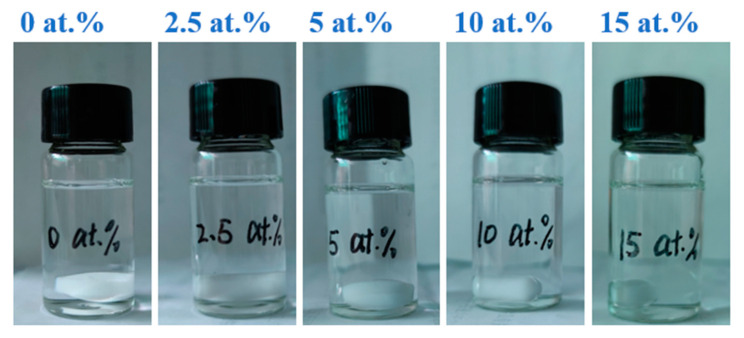
STO precursors with different Si concentrations.

**Figure 4 membranes-12-00590-f004:**
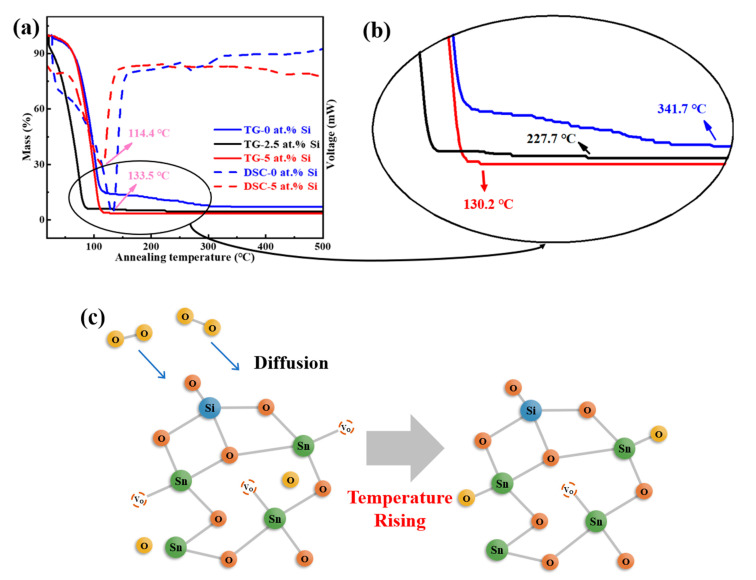
(**a**) DSC-TG curves of STO solutions with different Si concentrations and (**b**) local enlarged view of the TG. (**c**) Sketch map of the relationship among temperature, O and V_O_ in STO solidification process.

**Figure 5 membranes-12-00590-f005:**
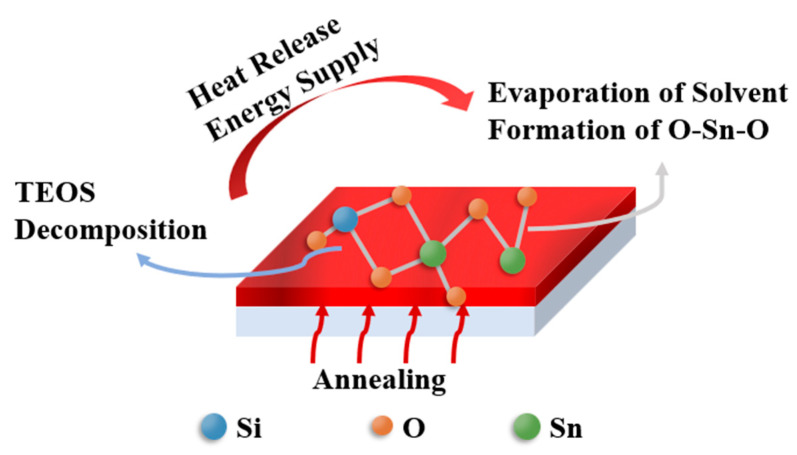
Sketch map of TEOS decomposition and SnOx synthesis.

**Figure 6 membranes-12-00590-f006:**
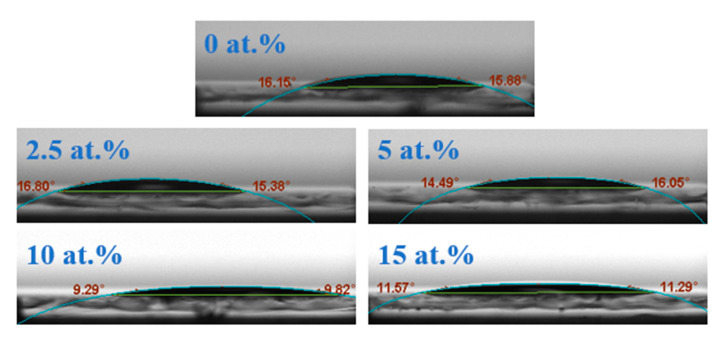
The contact angle of STO precursors on glass substrate.

**Figure 7 membranes-12-00590-f007:**
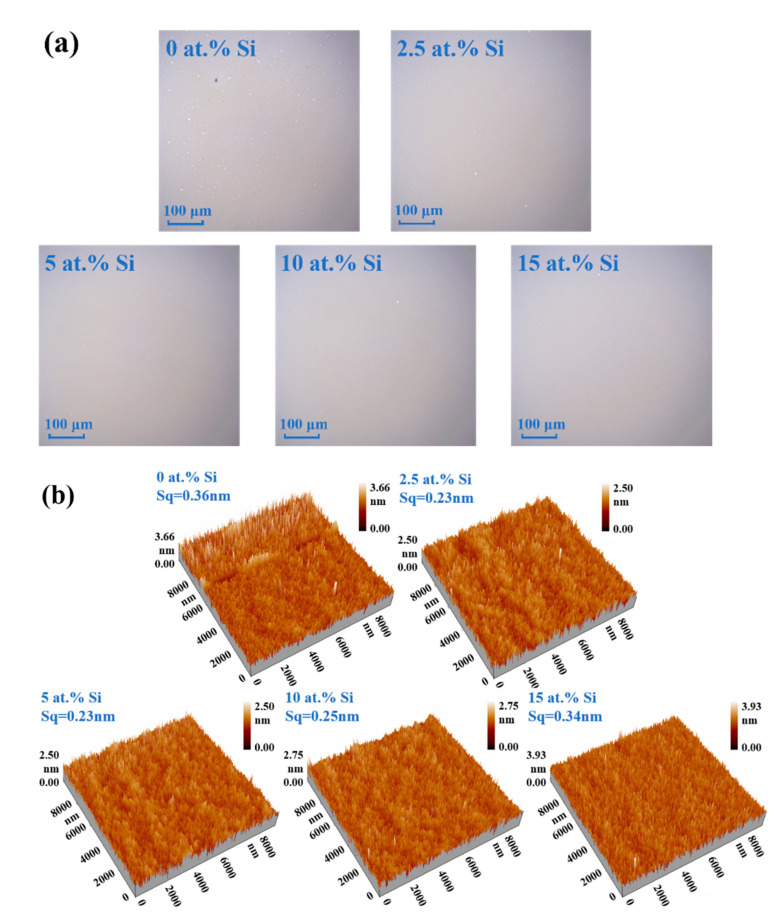
(**a**) LCSM images and (**b**) AFM 3D images (10 × 10 μm^2^) of STO thin films with different Si concentrations (300 °C).

**Figure 8 membranes-12-00590-f008:**
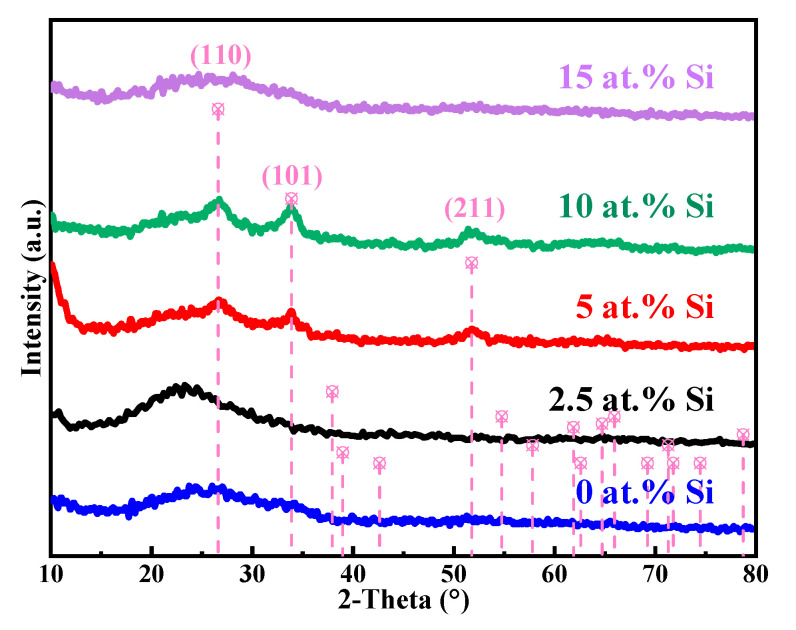
XRD patterns of STO films with different Si concentrations.

**Figure 9 membranes-12-00590-f009:**
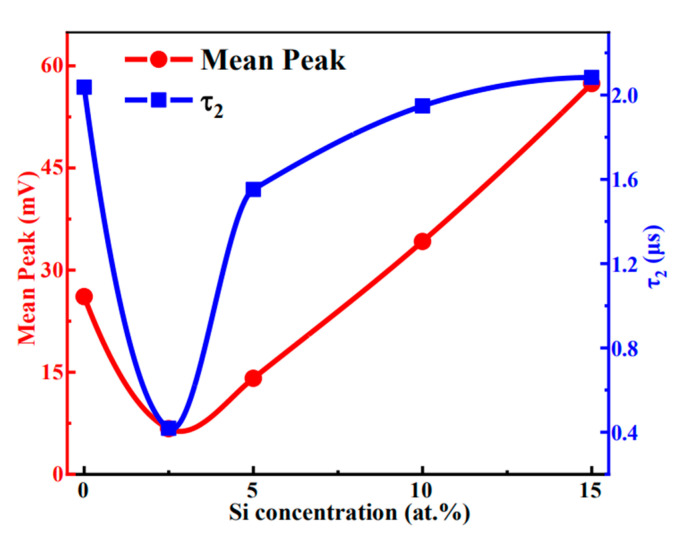
Mean peak and τ_2_ value of STO films as a function of Si concentration.

**Figure 10 membranes-12-00590-f010:**
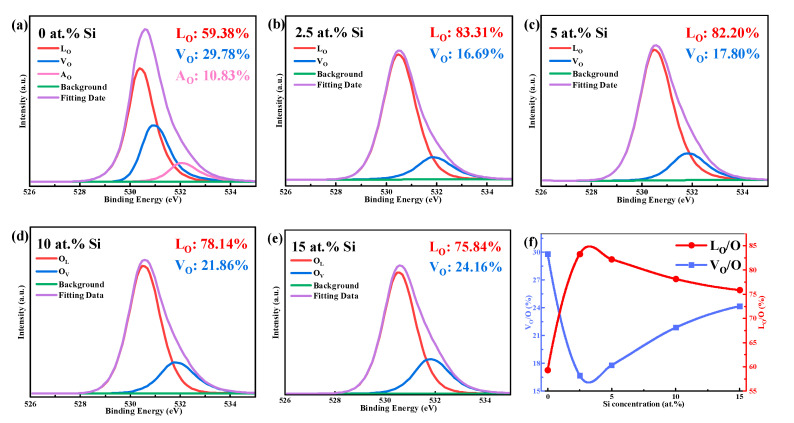
(**a**–**e**) XPS spectra for O 1s peaks of STO films with different Si concentrations. (**f**) Variation of L_O_/(L_O_ + V_O_ + A_O_) and V_O_/(L_O_ + V_O_ + A_O_) with Si concentration.

**Figure 11 membranes-12-00590-f011:**
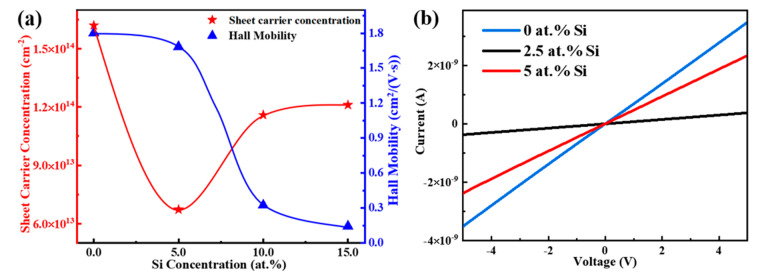
(**a**) Sheet carrier concentration and hall mobility of STO films with different Si concentrations. (**b**) Current-voltage response curves of STO films with different Si concentrations.

**Figure 12 membranes-12-00590-f012:**
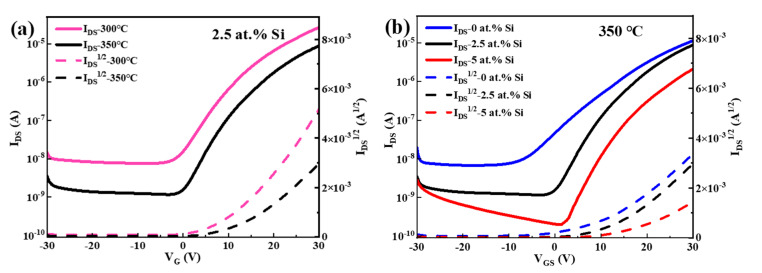
(**a**) Transfer characteristics of the STO TFTs with 2.5 at.% Si prepared at 300 °C and 350 °C. (**b**) Transfer characteristics of the STO TFTs with different Si concentrations annealed at 350 °C.

**Table 1 membranes-12-00590-t001:** Extracted performance parameters of 350 °C annealed STO TFTs.

Si (at.%)	I_on_ (A)	I_off_ (A)	I_on_/I_off_	V_on_ (V)	μ (cm^2^/(V·s))	SS (V/Dec)
0.0	1.04 × 10^−5^	6.76 × 10^−9^	1.54 × 10^3^	−17.22	0.05	8.73
2.5	8.84 × 10^−6^	1.19 × 10^−9^	7.43 × 10^3^	−2.00	0.32	4.24
5.0	2.10 × 10^−6^	2.01 × 10^−10^	1.04 × 10^4^	1.51	0.13	3.48

## Data Availability

Data are contained within the article.

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
