# Peer review of "Solution-Processed Silicon Doped Tin Oxide Thin Films and Thin-Film Transistors Based on Tetraethyl Orthosilicate"

_membranes, 2022, doi:10.3390/membranes12060590_

Round 1
Reviewer 1 Report
In this paper, authors presented STO-TFTs with TEOS as Si dopant and various doping level of Si were investigated to explore diverse properties. The work seems to be an average work. I have many points on this work but few comments are mentioned here.
1. The significance of the work is missing in the introduction part, authors should have mentioned why they carried out this work and what is actual novelty.
2. The effect of Si doping was monitored for investigating concentrations of oxygen vacancies, however the XPS results are not sufficient and authors need to provide more experimental evidences or cite more relevant papers such as; https://doi.org/10.1039/C4TC01984A,
https://doi.org/10.1016/j.mtcomm.2017.03.002,
3. The morphology results are not good too, why authors didn't use SEM or cross-sectional TEM for their films morphology?
4. The English level of the work need some improvements.
Author Response
Please see the attachment. Thank you for your letter and for the reviewers' comments concerning our manuscript entitled “Solution-Processed Silicon Doped Tin Oxide Thin Films and Thin-Film Transistors Based on Tetraethyl Orthosilicate”. Those comments are valuable and very helpful for revising and improving our paper.

Reviewer 2 Report
It is a very interesting paper with a few corrections to be done before publishing. Most of it, concerned to writing. My commnets are listed in the following.
1) In the phrase, "Notably, the electronic structure of Sn4+ (4d105s2) is similar to that of In3+ (4d105s0), making SnO2 and In2O3 own superior electrical properties in amorphous state ", the word "superior" should be better explained. What do the Authors mean by that?
2) Please, provide the reference from which "(~308 RMB/kg)," was obtained.
3) The phrase "Si is green, 60 non-toxic, and rich in resources. Si4+ has the same valence state as Sn4+ and will not intro-61 duce new charges into SnO2. Meanwhile, the binding energy of Si-O (799.6 kJ/mol) is 62 higher than that of Sn-O (531.8 kJ/mol), and the Lewis acid strength of Si (8.096) is also 63 significantly higher than that of Sn (1.617), which can suppress the formation of VO more 64 effectively [19], thus lowering the carrier concentration of SnO2 films and improving the 65 device performance." is probably one of the most important. Please, provide more references to that.
4) Please, rewrite "by stirring on a magnetic stirrer for 0.5 h to fully mixed." in Materials and Methods. The whole phrase is weird.
5) It is necessary to start the following phrase "And TEOS was added ..." with "And"?
6) In "and then spun coating by homoge-81 nizer at 5000 rpm for 30 s", use "spin coating" or "spun". What do you mean by "homogenizer"? Which parameter gets more homogeneous?
7) It is worth better explaining why did the Authors use "Nd" in the structure (e.g. "Al: Nd/Al2O3: Nd"). That could be done in the Introduction, depending on the length of the addition. Please, provide a reference on that.
8) Please, write "photoconductivity decay".
9) In Results, contact angle calculations provided values such as "16.15°". Did the Authors have this precision?
10) SEM and AFM are better techniques for what is shown in Figure 5.
11) In " However, for 15 145 at.% Si, the diffraction peaks disappear completely, which may be explained by that a 146 large amount of Si enters into SnO2 crystal and destroys its normal lattice structure, thus 147 suppressing the crystallization of SnO2.", is it acceptable to consider 15 at.% still doping?
12) In "The internal defects 151 of the film will significantly affect the carrier concentration of the film and the perfor-152 mance of STO-TFTs. ", the present tense instead of the future one would be more adequate. Even better, rewrite the phrase or reposition it in the paragraph.
13) In the same paragraph, "and when exceeding 10 at.%, the peak value of the STO film is 161 greater than that of pure SnO2 film. It reveals that high content of Si will increase the den-162 sity of shallow level defects in SnO2, which is unfavorable to reduce the carrier concentra-163 tion of SnO2 films.", the conclusions withdrawn from Figure 7 can be better written and explained to the reader. It seems incomplete. Please, deepen it.
14) Please, avoid using "etc" as in "Figure 8 displays the O 1s XPS spectra of STO thin films as a function of Si concen- tration. The O elements in SnO2 films mainly exist in the form of lattice oxygen (LO), ad- sorbed oxygen (AO), and VO, etc. ".
15) Please, use "addition" in " implying that Si additive ". The word "additive" appeared in other phrases too.
16) In the following phrase, "However, as the Si concentration rises from 2.5 at.% to 15 at.%, the VO ratio in STO films slightly increases but still lower than that of 0 at.% Si. It may be due to a disorder the structure when a large amount Si intercalated in the lattice [25], as indicated of LO ratio (Figure 8f).", there are minor errors almost everytwhere (I found at least four). Please, check it carefully and rewrite it.
17) Where is the experimental data in Figure 8?
18) At this point, considering the focus on electrical devices, I was asking myself why did the Authors choose Membranes journal. If it makes sense, it would be worth to somehow explain that in in the text (not directly to the name of the journal, but how does it relate to membranes in general).
19) The phrase "It demonstrates 200 that the STO films with 2.5 at.% Si are expected to improve the problem of turning off the 201 SnO2 based TFTs resulted from the high Ioff." can not be wirtten from Figure 9. It can be interpreted as an indication/direction. By the way, how was mobility obtained? Please, give equations and more details on that.
20) The text related to Figures 10 and 11 should be better placed in the beginning of the Results or before any electrical measurement.
21) In FIgure 12b, why not keep the color pattern for each doping used in previous Figures?
Author Response
Please see the attachment. Thank you for your letter and for the reviewers' comments concerning our manuscript entitled “Solution-Processed Silicon Doped Tin Oxide Thin Films and Thin-Film Transistors Based on Tetraethyl Orthosilicate”. Those comments are valuable and very helpful for revising and improving our paper. We have studied comments carefully and have made corrections which we hope meet with approval.
